# Risk Score Model for Microvascular Invasion in Hepatocellular Carcinoma: The Role of Tumor Burden and Alpha-Fetoprotein

**DOI:** 10.3390/cancers13174403

**Published:** 2021-08-31

**Authors:** Jin-Chiao Lee, Hao-Chien Hung, Yu-Chao Wang, Chih-Hsien Cheng, Tsung-Han Wu, Chen-Fang Lee, Ting-Jung Wu, Hong-Shiue Chou, Kun-Ming Chan, Wei-Chen Lee

**Affiliations:** Division of Liver and Transplantation Surgery, Department of General Surgery, Linkou Chang-Gung Memorial Hospital, Taoyuan City 333, Taiwan; b9302012@cgmh.org.tw (J.-C.L.); mp0616@cgmh.org.tw (H.-C.H.); b9002072@cgmh.org.tw (Y.-C.W.); chengcchj@cgmh.org.tw (C.-H.C.); domani@cgmh.org.tw (T.-H.W.); lee5310@cgmh.org.tw (C.-F.L.); wutj5056@cgmh.org.tw (T.-J.W.); chouhs@cgmh.org.tw (H.-S.C.); chankunming@cgmh.org.tw (K.-M.C.)

**Keywords:** microvascular invasion, hepatocellular carcinoma, risk score model, total tumor volume, alpha-fetoprotein

## Abstract

**Simple Summary:**

Microvascular invasion (MVI) is the most consistently reported risk factor for recurrence after curative treatment in hepatocellular carcinoma (HCC), but the preoperative prediction of MVI is still challenging. We retrospectively collected 1153 patients who underwent liver resection for HCC, and our multivariate analysis revealed preoperative total tumor volume (TTV) and alpha-fetoprotein (AFP) to be independent risk factors for MVI. We used both factors to build a risk score model that is easy to calculate and objective, with minimal user bias. The preoperative prediction of MVI can guide the treatment plan of HCC, including surgical planning, criteria for transplantation, and adjuvant or neoadjuvant therapy. Our risk score model is easily and widely applicable with moderate performance, which optimizes clinical practice and helps study design in the future.

**Abstract:**

Microvascular invasion (MVI) is a significant risk factor for the recurrence of hepatocellular carcinoma, but it is a histological feature that needs to be confirmed after hepatectomy or liver transplantation. The preoperative prediction of MVI can optimize the treatment plan of HCC, but an easy and widely applicable model is still lacking. The aim of our study was to predict the risk of MVI using objective preoperative factors. We retrospectively collected 1153 patients who underwent liver resection for HCC, and MVI was found to be associated with significantly poor disease-free survival. The patients were randomly split in a 3:1 ratio into training (n = 864) and validation (n = 289) datasets. The multivariate analysis of the training dataset found preoperative total tumor volume (TTV) and alpha-fetoprotein (AFP) to be independent risk factors for MVI. We built a risk score model with cutoff points of TTV at 30, 60, and 300 cm^3^ and AFP at 160 and 2000 ng/mL, and the model stratified the risk of MVI into low risk (14.1%), intermediate risk (36.4%), and high risk (60.5%). The validation of the risk score model with the validation dataset showed moderate performance (the concordance statistic: 0.731). The model comprised simple and objective preoperative factors with good applicability, which can help to guide treatment plans for HCC and future study design.

## 1. Introduction

Hepatocellular carcinoma (HCC) is the most common primary malignancy of the liver and one of the leading cancers for both males and females in Taiwan because of the high prevalence of chronic hepatitis B (HBV) and hepatitis C (HCV) viral infection [1]. Liver resection is the main curative treatment for resectable HCC, but the five-year recurrence rate is not satisfactory, with reported rates between 58.4 and 100% [2,3,4,5,6].

Microvascular invasion (MVI) is the most consistently reported risk factor for the post-hepatectomy recurrence of HCC [5,7], and the incidence of MVI is around 15–57.1%, as reported by a systemic review that included 20 studies [8]. MVI has been proven to be an independent risk factor for HCC [9,10,11] and is a cause for poor survival outcome, even in small HCCs [12] or after propensity score matching [13]. Several studies have found MVI to be an independent risk factor for a higher recurrence rate and increased mortality in HCC patients who have also received liver transplantation [14,15,16,17]. However, MVI is a histological feature that needs to be confirmed after hepatectomy or liver transplantation.

As an important prognostic factor, preoperatively predicting MVI optimizes the decision of treatment for HCC. Several studies have attempted to predict MVI using preoperative factors. The incidence of MVI in hepatectomy or transplantation specimens has been found to increase with tumor size, the presence of multiple tumors, a high AFP level, and a high histological grade [12,18,19]. Risk score models and nomogram systems have been built using preoperative clinical, laboratory, and image factors to more precisely predict MVI [20,21,22]. However, the models from previous studies are only applicable to certain subsets of patients based on the number of tumors and their size or viral hepatitis status, and these models are often affected by subjective assessment (typical image pattern or complete tumor capsule) or CT radiomic factors. To overcome these issues, we retrospectively enrolled 1153 patients with HCC who underwent hepatectomy to build a prediction model for MVI using simple and objective preoperative factors.

## 2. Materials and Methods

### 2.1. Patients

From January 2003 to December 2012, 1531 consecutive patients underwent liver resection for HCC at the Department of General Surgery, Chang-Gung Memorial Hospital, and the pathological reports proved the diagnosis of HCC. Patients with the following conditions were excluded from our study: the presence of other malignancies, radiofrequency ablation (RFA), transcatheter arterial chemoembolization (TACE) or radiotherapy used to treat HCC before liver resection, positive resection margin, macrovascular invasion proven by pathologic report, TNM stage IV, or missing laboratory data (Figure 1). In the end, 1153 patients were enrolled in our study, which was approved by the local ethics committee of Chang-Gung Memorial Hospital (Institutional. Review Board No. 104-3900B).

### 2.2. Treatment Strategy and Liver Resection

Our treatment strategy followed the Barcelona Clinic Liver Cancer (BCLC) staging system and treatment strategy. Liver resection is the choice of treatment for very early stage and early stage solitary HCC with preserved liver function assessed by Child–Pugh classification and the indocyanine green (ICG) test. Intraoperative sonography was used to determine the resection route. For inflow control of liver during operation, intermittent Pringle’s maneuver (15 min of clamping followed by 5 min of release) was applied in selected cases. We used an ultrasonic dissector or a pean-clamp for the transection of the liver parenchyma.

### 2.3. Clinicopathological Profiles and Definitions

We collected patients’ clinical information via preoperative computer tomography (CT) imaging, laboratory examination (including hematology, biochemistry, tumor marker alpha-fetoprotein (AFP), and hepatitis serology tests), surgical features, pathologic features, tumor recurrence, and details of the last follow-up or date of death from the medical charts and the Taiwan Cancer Registry. The number of tumors and their sizes (length and width) were measured from pre-operative CT. The total tumor volume (TTV) was the sum of the volume of every tumor, which was calculated as: length × width × width × 0.52, which is a standard way to determine the tumor volume of nude mice in the laboratory [23,24]. The cut-off points for TTV (30, 60, and 300 cm^3^) and AFP (160 and 2000 ng/mL) were decided with a classification tree. The definition of MVI is the presence of tumor cells inside in the endothelial vascular lumen under microscopy only. The diagnosis of tumor recurrence was made with dynamic CT imaging.

### 2.4. Random Split of Patients and Construct the Risk Score Model

We split our dataset with the shuffle split method in a 3:1 ratio into a training dataset (75%; n = 864) and a validation dataset (25%; n = 289) (Figure 1). The risk score model was developed by the training dataset using the significant risk factors from the multivariate analysis. The risk points of each factor were decided based on its regression coefficients. The risk score model was validated with the validation dataset.

### 2.5. Biostatistics

Preoperative clinical and image characteristics were analyzed using multivariate analysis to identify the independent risk factors of microvascular invasion. All statistical analyses were performed using IBM SPSS 22 (SPSS, Inc., Chicago, IL, USA) software. Baseline characteristics were compared using the chi-square test for categorical variables and analysis of variance for continuous variables. The disease-free survival rate was calculated using the Kaplan–Meier method and compared using log-rank tests. Statistical significance was set at *p* < 0.05.

## 3. Results

### 3.1. Prognostic Value of Microvascular Invasion

Before the partition of the dataset (n = 1153), the presence of MVI was confirmed to be a significant poor prognostic factor for disease-free survival using the Kaplan–Meier method. The 1-, 3-, and 5-year disease-free survival rates in patients with MVI were 57.2%, 36.8%, and 29.4%, respectively; the same in patients without MVI were 81.2%, 59.2%, and 46.3%, respectively. The *p*-value was less than 0.001 (Figure 2).

### 3.2. Characteristics of the Patients

After selection by our exclusion criteria, 1153 HCC patients who underwent liver resection were enrolled in this study. The mean age of the patients was 58.7 ± 12.8 years old, and 78.5% of them were male. Most of the patients had HBV infection (52.6%), HCV infection (24.1%), or HBV and HCV infection (6.5%), and 97.6% of the patients’ liver function were classified as Child’s A. In preoperative CT, most of the patients had a single tumor (87.5%), and 10.1% of them had two tumors. Only 2.4% of the patients were found to have had >3 tumors in the preoperative CT imaging. The median TTV of the tumors calculated based on preoperative CT was 17.6 cm^3^ (interquartile range: 5.6–78.6 cm^3^). The 1-, 3-, and 5-year disease-free survival rates were 75.3%, 42.2%, and 29.7%, respectively; the 1-, 3-, and 5-year overall survival rates were 94.0%, 70.2%, and 51.2%, respectively. Most of the recurrence had only intrahepatic recurrence (91.3%). There was no significant difference between the training dataset and the validation dataset in all the preoperative characteristics and the incidence of MVI (24.1% and 24.2%, respectively; Table 1).

### 3.3. Prognostic Factors for Microvascular Invasion in Training Dataset

The univariate analysis of the training dataset identified high AFP (160–2000 and >2000 ng/mL), high TTV (30–60, 60–300, and >300 cm^3^), and multiple tumor (>3) as significant risk factors for microvascular invasion. Multivariate analysis using the significant factors identified in univariate analysis found only high AFP (160–2000 and >2000 ng/mL) and high TTV (30–60, 60–300, and >300 cm^3^) to be independent risk factors for microvascular invasion (Table 2).

### 3.4. Construction and Validation of the Risk Score Model for Microvascular Invasion

Score points were given to all the independent risk factors according to their regression coefficients, as shown in Table 3. AFP values between 160 and 2000 ng/mL and over 2000 ng/mL were given 1 and 3 points, respectively, and TTV values between 30 and 60 cm^3^, between 60 and 300 cm^3^, and over 300 cm^3^ were given one, two, and three points, respectively. There were six total points. In each score from 0 to 6, the incidences of MVI were 13.0%, 16.1%, 27.8%, 46.8%, 62.1%, 56.3%, and 64.0%, respectively, and the *p*-value was less than 0.001 (Table 3). The risk of MVI was stratified to low risk (score 0–1; MVI probability of 14.1%), intermediate risk (score 2–3; MVI probability of 36.4%), and high risk (score 4–6; MVI probability of 60.5%). The area under curve (AUC) of the receiver operating characteristic (ROC) curve was 0.714, and the calibration plot showed good performance of the prediction model, with a low mean absolute error at 0.023 (Figure 3a and Figure 4). A validation dataset was used to validate the risk score model, and the AUC of the ROC curve was found to be 0.731. (Figure 3b) The values of Nagelkerke’s R2 and the results of the Hosmer–Lemeshow test showed that the overall model fit was good, with a median effect size (Table 4).

## 4. Discussion

MVI is the most consistently reported risk factor for HCC recurrence after liver resection [5,7] or liver transplantation [14,15,16,17], but it can only be confirmed by a pathologist after surgical treatment. The results of our study found TTV and AFP levels to be independent predictors for MVI, which can thus be used to build an effective risk score model.

The authors of several past studies have tried to preoperatively predict MVI in HCC by using clinical, laboratory, and image factors. Tumor size is the most frequently mentioned predictor for MVI. Pawlik et al. [19] found that the incidence of MVI increases with increased tumor size. AFP level, histological grade, and the presence of multiple tumors have also been found to be common predictors for MVI [12,25]. Using ultrasound CT, MRI, and PET, multiple image features, including irregular tumor margin, incomplete capsule, typical dynamical pattern, low apparent diffusion coefficient (ADC) values and peri-tumor hypo-intensity in MRI, and a high tumor/liver activity ratio (RSUVmax) in PET, have been found to correlate with MVI [26,27]. Liver imaging reporting and data system (LI-RADS) 5 subclass in LI-RADS classification was found to associated with a higher frequency of microvascular invasion [28]. A risk score model was built for only multinodular HCC using AFP, gamma glutamyl transpeptidase, tumor size over 8 cm, and tumor number over 3 [20]. Several nomogram models have been set up with both clinical and image predictors, but most of them comprised subjective image factors or technique-dependent radiomic factors that increased the difficulty and bias in clinical application [21,22,29,30]. Cucchetti et al. [31] used a machine learning method (artificial neural network) to build a prediction model for MVI and tumor grades in a pilot study with a small sample size (250 patients with both resection and transplantation).

TTV, which is a combination of the tumor size and the number of tumors to reflect the tumor burden, is a strong prognostic factor for HCC recurrence after liver resection [32] or liver transplantation [33]. AFP level is a biomarker of HCC, and a high AFP level is a significant risk factor for tumor recurrence after curative treatment [34,35]. AFP level has also been proven to be associated with the risk of MVI and tumor differentiation [36]. TTV and AFP have been used to build an expanded criteria (TTV < 115 cm^3^ and AFP < 400 ng/mL with at least eight months of waiting time) for liver transplantation with satisfactory post-transplant survival [37].

In our study, multivariate analysis revealed that preoperative TTV and AFP were re independent risk factors for MVI. Most of our patients had a HBV infection (52.6%) because of the high prevalence of chronic HBV infection in Taiwan, but the hepatitis status was not found to be a risk factor for developing microvascular invasion in univariate analysis. A risk score model for the prediction of MVI was built with preoperative TTV and AFP, which represent the tumor burden and biology, respectively. These two factors, with cut-points selected with a classification tree, are easy to calculate and are objective, with minimal user bias. These characteristics make the prediction model more applicable for all subspecialists, and the model is not limited by a patient’s viral hepatitis status or the size or number of tumors. The validation of our prediction model using a validation dataset created by the shuffle split method showed moderate performance, with an AUC of 0.731. The values of Nagelkerke R2 and the results of the Hosmer–Lemeshow test showed that the overall model fit was good, with a median effect size.

The preoperative prediction of MVI can optimize the treatment plan for HCC. Han et al. [38] found that concomitantly having a narrow resection margin and positive MVI increased the risk of postoperative recurrence after liver resection for HCC, and a wider resection margin was recommended if MVI could be predicted before liver resection. A systemic review and meta-analysis study found that anatomic resection provided better disease-free survival and overall survival compared to non-anatomic resection in patients with MVI [39]. Liver transplantation should be preferred over liver resection in patients with a high risk of MVI to remove possible intrahepatic micro-metastasis. The risk of MVI can be incorporated into the expanded criteria for liver transplantation to preoperatively evaluate the benefit of transplantation. Thus, preoperatively being able to determine MVI is of utmost importance to a surgeon. With the rapid development of immunotherapy in HCC, MVI is also a possible indicator of adjuvant therapy or neoadjuvant therapy with immunotherapy for HCC in the future.

There were several limitations of our study. It was a retrospective study, and it may have had selection bias caused by missing laboratory data (most of the missing TTV data were from low-risk patients with very small tumors). We only conducted an internal validation of the risk score model using the shuffle split method, and this model needs external validation to examine its transferability to different populations.

## 5. Conclusions

MVI is a significant poor prognostic factor for HCC recurrence. We built a risk score model to preoperatively predict MVI using TTV and AFP, which represent the tumor burden and biology, respectively. The risk of MVI was stratified into low, intermediate, and high risk (14.1%, 36.4%, and 60.5%, respectively, risk of developing MVI). By using this easily and widely applicable model, the treatment plan for HCC can be adjusted based on the risk of MVI to optimize survival outcomes.

## Figures and Tables

**Figure 1 cancers-13-04403-f001:**
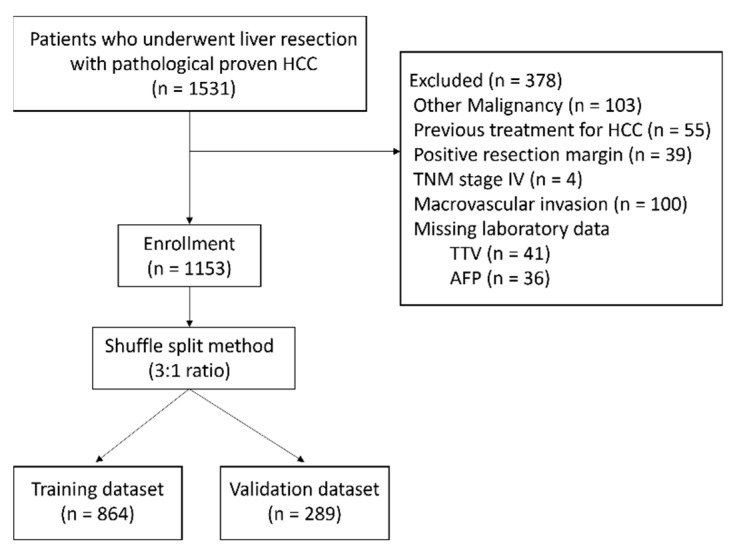
The CONSORT flow diagram of the enrollment of patients and the random split.

**Figure 2 cancers-13-04403-f002:**
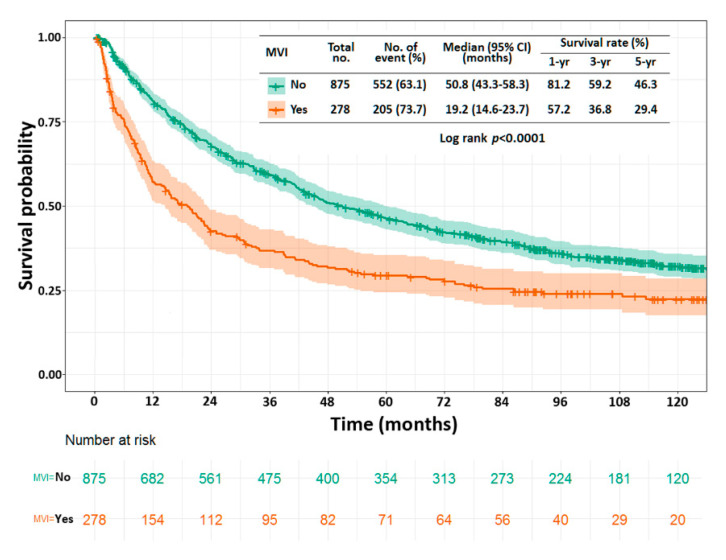
Kaplan–Meier curve for disease-free survival in patients with (red line) or without (green line) microvascular invasion.

**Figure 3 cancers-13-04403-f003:**
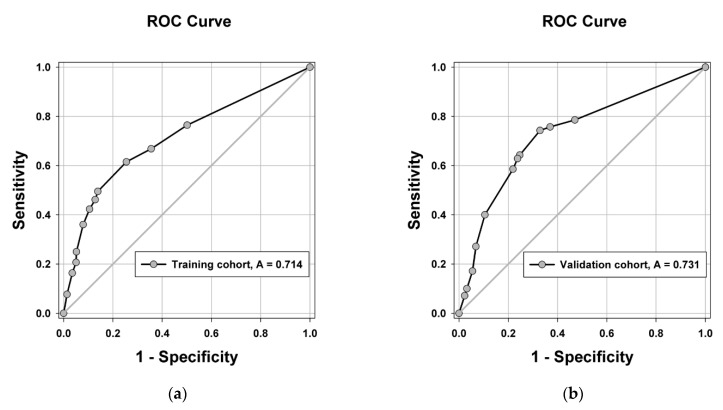
Receiver operating characteristic (ROC) curve of the risk score model for the prediction of microvascular invasion in (**a**) a training dataset with an area under the curve (AUC) of 0.714 and (**b**) a validation dataset with an AUC of 0.731.

**Figure 4 cancers-13-04403-f004:**
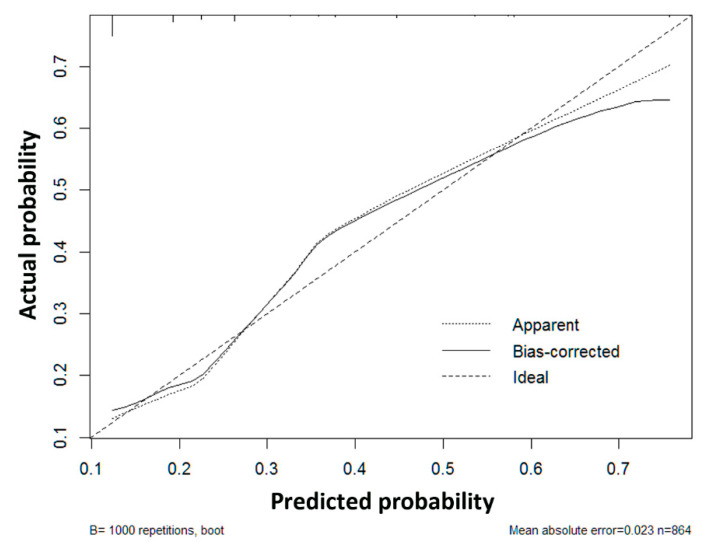
Calibration plot of actual versus predicted probability of microvascular invasion in a training dataset with a mean absolute error of 0.023.

**Table 1 cancers-13-04403-t001:** Clinical characteristics of patients classified into training and validation datasets.

Factors	Total (n = 1153)	Training Dataset (n = 864)	Validation Dataset (n = 289)	*p*-Value
Age (years)	58.7 ± 12.8	58.4 ± 13.1	59.8 ± 11.5	0.101
Sex				0.761
Male	905 (78.5)	680 (78.7)	225 (77.9)	
Female	248 (21.5)	184 (21.3)	64 (22.1)	
Viral hepatitis status				0.349
HBV	606 (52.6)	458 (53.0)	148 (51.2)	
HCV	278 (24.1)	207 (24.0)	71 (24.6)	
HBV + HCV	75 (6.5)	61 (7.0)	14 (4.8)	
NBNC	194 (16.8)	138 (16.0)	56 (19.4)	
Albumin (g/mL)	4.15 ± 0.45	4.15 ± 0.45	4.14 ± 0.45	0.940
Platelet count (1000/uL)	177.8 ± 73.9	176.8 ± 71.0	180.7 ± 82.1	0.450
Bilirubin (mg/dL)	0.82 ± 0.97	0.85 ± 1.10	0.76 ± 0.40	0.199
INR	1.08 ± 0.09	1.08 ± 0.09	1.08 ± 0.10	0.560
ICG (%)				0.138
≤10	728 (66.7)	556 (67.9)	172 (63.0)	
>10	364 (33.3)	263 (32.1)	101 (37.0)	
Child classification				0.469
A	1125 (98.7)	842 (98.8)	283 (98.3)	
B	15 (1.3)	10 (1.2)	5 (1.7)	
AFP (ng/mL) *	19.3 (5.5–276.6)	19.4 (5.6–318.7)	18.9 (5.1–168.6)	0.212
≤160	805 (69.8)	595 (68.9)	210 (72.7)	0.473
>160, ≤2000	231 (20.0)	179 (20.7)	52 (18.0)	
>2000	117 (10.2)	90 (10.4)	27 (9.3)	
Total tumor volume (cm^3^) *	17.6 (5.6–78.6)	17.3 (5.5–77.2)	18.6 (5.8–90.3)	0.448
≤30	681 (59.1)	515 (59.6)	166 (57.4)	0.917
>30, ≤60	135 (11.7)	101 (11.7)	34 (11.8)	
>60, ≤300	221 (19.2)	163 (18.9)	58 (20.1)	
>300	116 (10.1)	85 (9.8)	31 (10.7)	
Tumor number				0.346
1	1009 (87.5)	756 (87.5)	253 (87.5)	
2	116 (10.1)	90 (10.4)	26 (9.0)	
≥3	28 (2.4)	18 (2.1)	10 (3.5)	
Microvascular invasion				0.960
Yes	278 (24.1)	208 (24.1)	70 (24.2)	
No	875 (75.9)	656 (75.9)	219 (75.8)	
Disease-free survival (%)				0.501
1 year	75.3	74.9	76.6	
5 years	42.2	42.9	40.4	
10 years	29.7	29.9	29.2	
Overall survival (%)				0.403
1 year	94.0	93.3	96.2	
5 years	70.2	71.0	67.7	
10 years	51.2	51.7	49.8	
Recurrence pattern				1.000
Intrahepatic recurrence only	691 (91.3)	513 (91.3)	178 (91.3)	
Extrahepatic recurrence	66 (8.7)	49 (8.7)	17 (8.7)	

Figures are numbers with percentages in parentheses unless otherwise stated. Continuous variables are presented with mean ± standard deviation. * Continuous variables with outliers are presented with median (interquartile range).

**Table 2 cancers-13-04403-t002:** Prognostic factors for microvascular invasion according to univariate and multivariate analysis using a training dataset.

Factors	Odds Ratio	95% C.I.	*p*-Value	Odds Ratio	95% C.I.	*p*-Value
Age (years)						
≤60/>60	1.333	0.972–1.826	0.074			
Gender						
Male/Female	1.181	0.799–1.747	0.404			
Hepatitis						
HBV/NBNC	0.958	0.632–1.474	0.846			
HCV/HBNC	0.654	0.392–1.090	0.103			
HBV and HCV/NBNC	0.669	0.321–1.394	0.283			
Albumin (g/dL)						
≤3.5/>3.5	1.146	0.661–1.986	0.627			
Platelet count (1000/uL)						
>150/≤150	1.281	0.923–1.779	0.139			
ICG (R15)						
>10/≤10	1.131	0.798–1.601	0.490			
Child classification						
B/A	3.189	0.914–11.127	0.069			
AFP (ng/mL)						
>160, <2000/≤160	1.700	1.152–2.508	0.008	1.658	1.103–2.491	0.015
>2000/≤160	5.897	3.700–9.398	<0.001	4.030	2.452–6.625	<0.001
Total tumor volume (cm^3^)						
>30, ≤60/≤30	2.075	1.255–3.432	0.004	1.983	1.181–3.329	0.010
>60, ≤300/≤30	2.977	1.988–4.459	<0.001	2.404	1.573–3.673	<0.001
>300/≤30	7.379	4.510–12.075	<0.001	5.335	3.177–8.960	<0.001
Tumor number						
2/1	0.996	0.594–1.669	0.987	0.908	0.518–1.590	0.735
≥3/1	4.089	1.590–10.518	0.003	2.505	0.917–6.844	0.073

**Table 3 cancers-13-04403-t003:** Risk score model for the prediction of microvascular invasion.

Predictor Variables	Regression Coefficients (β)		Categories		Point		
Total tumor volume	1.730		>300		3		
	0.913		>60, ≤300		2		
	0.708		>30, ≤60		1		
			≤30 *		0		
AFP	1.369		>2000		3		
	0.526		>160, ≤2000		1		
			≤160 *		0		
Total Score	0	1	2	3	4	5	6
Probability of MVI	13.0% (49/376)	16.1% (31/193)	27.8% (32/115)	46.8% (44/94)	62.1% (18/29)	56.3% (18/32)	64.0% (16/25)
Risk	Low risk		Intermediate risk		High risk		
Probability of MVI	14.1%		36.4%		60.5%		

* Reference category.

**Table 4 cancers-13-04403-t004:** The comparison of performance measures by using logistic regression mode in MVI probability.

Model	Overall Performance Measure	Discrimination	Calibration
	Nagelkerke R^2^	C Statistic	Hosmer–Lemeshow Test
Training cohort	0.174	0.714	0.314
Validation cohort	0.193	0.731	0.342

## Data Availability

The data presented in this study are available on request from the corresponding author.

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
