# Peer review of "Risk Score Model for Microvascular Invasion in Hepatocellular Carcinoma: The Role of Tumor Burden and Alpha-Fetoprotein"

_cancers, 2021, doi:10.3390/cancers13174403_

Round 1

Reviewer 1 Report

This retrospective paper is very interesting to describe a predictive model of recurrent of hepatocellular carcinoma. However, the objective of the paper is not clearly defined either in the abstract or at the end of the introduction. The conclusions do not clearly and briefly explain the results expressed in table3.

Author Response

Point 1: This retrospective paper is very interesting to describe a predictive model of recurrent of hepatocellular carcinoma. However, the objective of the paper is not clearly defined either in the abstract or at the end of the introduction. The conclusions do not clearly and briefly explain the results expressed in table3.

Response 1: Thank you for the important advice. The aim of our study was to build an objective and simple risk prediction model for microvascular invasion using preoperative factors because of the lack of an easy and widely applicable prediction method now. We have added the aim of our study in the abstract and at the end of the introduction. We have also added a brief explanation of the result of our risk stratification (Table 3) in the conclusion.

Reviewer 2 Report

In the present manuscript, Lee and collegues developed an easy-to-use model based on total tumor volume and AFP levels for the prediction of microvascular invasion (MVI) in patients with resectable HCC. The study cohort in terms of sample size is considerable and the manuscript is well written. Some minor concerns need to be addressed before final acceptance of the manuscipt.

1) Statistical analysis. Was data normality checked? Which statistical test was used? According to data distribution, please report continuous variable as mean +- SD or median (IQR) and use parametric or non-parametric test for analysis. For instance, variables such albumin and platelets rarely have a normal distribution...

2) Table 1. Please report AFP and total tumor volume also as continuous variable.

3) The accuracy of the model in the validation cohort corresponds to 0.731. I would say that this a moderate rather than a good accuracy (see abstract line 28, discussion line 209).

4) Discussion section. Does this study have some limitations? Please add a "limitations of the study" paragraph at the end of the discussion section.

Author Response

Point 1: Statistical analysis. Was data normality checked? Which statistical test was used? According to data distribution, please report continuous variable as mean +- SD or median (IQR) and use parametric or non-parametric test for analysis. For instance, variables such albumin and platelets rarely have a normal distribution...

Response 1: Thank you for the important comments. In the probability theory, the central limit theorem establishes that the distribution of the sample mean satisfies the normal distribution when the number of samples is above 30. Based on the relatively large study population (n = 1153), we used parametric test for most of the continuous variable. However, for AFP and TTV, there were many outliers with extremely high value, which affected their mean values so that the means were not representative of the dataset. So, for continuous variables with outliers, we presented them with median (interquartile range (IQR)) and used non-parametric test for analysis as suggested by the reviewer.

Point 2: Table 1. Please report AFP and total tumor volume also as continuous variable.

Response 2: Thank you for the helpful advice. We have added the presentation of AFP and TTV as continuous variables with median (IQR) in Table 1.

Point 3: The accuracy of the model in the validation cohort corresponds to 0.731. I would say that this a moderate rather than a good accuracy (see abstract line 28, discussion line 209).

Response 3: Thank you for your constructive comment to help give a better definition of the accuracy of our model. We have made the changes as advised by the reviewer.

Point 4: Discussion section. Does this study have some limitations? Please add a "limitations of the study" paragraph at the end of the discussion section.

Response 4: Thank you for pointing out this important aspect that we missed. We have added a paragraph to describe our limitations at the end of the discussion which is as follows: There were several limitations of our study. It was a retrospective study, and it may have had selection bias caused by missing laboratory data (most of the missing TTV data were from low-risk patients with very small tumors). We only conducted an internal validation of the risk score model using the shuffle split method, and this model needs external validation to examine its transferability to different populations.

Reviewer 3 Report

In their paper entitled "Risk score model for microvascular invasion in hepatocellular carcinoma: the role of tumor burden and alpha-fetoprotein" Lee and colleagues reported a retrospective single-center analysis of the predictors for MVI in HCC

Although the subject has been already analysed in several previous papers, the numerous population and the study design enhance the general interest of this research

I have few suggestions to improve the paper quality:

1) include a CONSORT diagram for patient selection

2) better define which BCLC classification was used

3) I am afraid that the proposed aFP splitting is not clinically useful: a value of 160 is already 10x the upper limit, while values >2000 are extremely high; a continuous evaluation should be more useful and also more rigorous form a statistical point of view

4) please provide the recurrence site (intrahepatic or extrahepatic), the recurrence free survival and overall survival in Table 1

5) Please add some comment on the high prevalence of HBV in the study population and its potential effect on the analysis

6) A recent study pointed out how LI-RADS categorization could represent another valuable predictor of MVI, and should be included in the discussion section: 10.3390/cancers13071671

7) Last, I suggest the Authors another round of English editing and restyling to match the 

Congratulation to the Authors for their interesting research

Best regards

Author Response

Point 1: include a CONSORT diagram for patient selection

Response 1: Thank you for the constructive comment. As suggested, we have added a CONSORT flow diagram to present the patient exclusion, enrollment and the split of dataset in Figure 1.

Point 2: better define which BCLC classification was used

Response 2: Thank you for the important comment. Our treatment strategy followed the Barcelona Clinic Liver Cancer (BCLC) staging system and treatment strategy. Liver resection is the choice of treatment for very early stage and early stage solitary HCC with preserved liver function assessed by the Child–Pugh classification and the indocyanine green (ICG) test. The above description has been added in the “2.2. Treatment Strategy and Liver Resection”.

Point 3: I am afraid that the proposed aFP splitting is not clinically useful: a value of 160 is already 10x the upper limit, while values >2000 are extremely high; a continuous evaluation should be more useful and also more rigorous form a statistical point of view

Response 3: Thank you for the helpful advice. The cut-off points (160 and 2000 ng/dL) in our study were defined by classification and regression tree. AFP values of 200 or 400ng/mL are the common cut-off points used in the diagnosis of HCC and prediction of disease prognosis. The value of 160ng/mL is close to the common cut-off value at 200ng/mL. One of the important problems of AFP value in statistics and clinical use is the presence of outliers. The highest AFP level in our study population is 2517987 ng/mL, and 16 patients have AFP level over 100000 ng/mL. The outliers affected the mean value of AFP and caused obvious bias when we analysed AFP as continuous variables. To overcome this problem, we used the categorical analysis to subgroup the outliers into the group with extremely high AFP value (> 2000ng/mL).

Point 4: please provide the recurrence site (intrahepatic or extrahepatic), the recurrence free survival and overall survival in Table 1

Response 4: Thank you for the important advice. As suggested, we have added the 1, 3 and 5-year disease-free survival, overall survival and the pattern of initial recurrence (intrahepatic recurrence only or presence of distant metastasis) in Table 1, and the results are described in “3.2. Characteristics of the Patients” as following: The 1-, 3-, and 5-year disease-free survival rates were 75.3%, 42.2%, and 29.7%, respectively; the 1-, 3-, and 5-year overall survival rates were 94.0%, 70.2%, and 51.2%, respectively. Most of the recurrence had only intrahepatic recurrence (91.3%).   

Point 5: Please add some comment on the high prevalence of HBV in the study population and its potential effect on the analysis

Response 5: Thank you for the important comment. In Taiwan, the prevalence of HBV infection was approximately 11-20% before the launch of the universal hepatitis B vaccination for infants in 1984, which was the highest in the world. Despite the success of universal vaccination, chronic HBV infection is still an important health problem in Taiwan. In our univariate analysis, hepatitis status (HBV, HCV or HBV + HCV) was not a risk factor for developing microvascular invasion. We described the results in the fourth paragraph of “Discussion”.

Point 6: A recent study pointed out how LI-RADS categorization could represent another valuable predictor of MVI, and should be included in the discussion section: 10.3390/cancers13071671

Response 6: Thank you for the helpful advice. We have added the reference to the second paragraph of “Discussion” as following: Liver imaging reporting and data system (LI-RADS) 5 subclass in the LI-RADS classification was found to associated with a higher frequency of microvascular invasion.

Point 7: Last, I suggest the Authors another round of English editing and restyling to match the

Response 7: Thank you for the important advice. We have submitted our manuscript to MDPI English editing, and the certification is attached as below.

Round 2

Reviewer 3 Report

The Authors properly addressed all the issues that were raised in the previous report

The revised version has been completely  refined by the english editing team, significantly improvine the paper readability

I think the paper is now suitable  for publication